# Statistical segmentation model for accurate electrode positioning in Parkinson's deep brain stimulation based on clinical low-resolution image data and electrophysiology

**Igor Varga** [1,2]*, **Eduard Bakstein** [1,3], **Greydon Gilmore** [4], **Jaromir May** [5], **Daniel Novak** [1]

**1** Department of Cybernetics, Czech Technical University in Prague, Prague, Czech Republic, **2** Czech Centre for Phenogenomics, Institute of Molecular Genetics of the Czech Academy of Sciences, Prague, Czech Republic, **3** National Institute of Mental Health, Klecany, Czech Republic, **4** Movement Disorder Centre, University Hospital, University of Western Ontario, Ontario, Canada, **5** Department of Neurosurgery, Na Homolce Hospital, Prague, Czech Republic

* varhaiho@cvut.cz, ivarhaizh@gmail.com

**Data Availability Statement:** The study data is owned by Lawson Health Research Institute, London, ON, Canada (Study REB #109045, ReDA

## Abstract

### Background

Deep Brain Stimulation (DBS), applying chronic electrical stimulation of subcortical structures, is a clinical intervention applied in major neurologic disorders. In order to achieve a good clinical effect, accurate electrode placement is necessary. The primary localisation is typically based on presurgical MRI imaging, often followed by intra-operative electrophysiology recording to increase the accuracy and to compensate for brain shift, especially in cases where the surgical target is small, and there is low contrast: e.g., in Parkinson's disease (PD) and in its common target, the subthalamic nucleus (STN).

### Methods

We propose a novel, fully automatic method for intra-operative surgical navigation. First, the surgical target is segmented in presurgical MRI images using a statistical shape-intensity model. Next, automated alignment with intra-operatively recorded microelectrode recordings is performed using a probabilistic model of STN electrophysiology. We apply the method to a dataset of 120 PD patients with clinical T2 1.5T images, of which 48 also had available microelectrode recordings (MER).

### Results

The proposed segmentation method achieved STN segmentation accuracy around dice = 0.60 compared to manual segmentation. This is comparable to the state-of-the-art on low-resolution clinical MRI data. When combined with electrophysiology-based alignment, we achieved an accuracy of 0.85 for correctly including recording sites of STN-labelled MERs in the final STN volume.

#2503, P.I. Prof. Mandar Jog). Sharing the de-identified study data requires signing a written data sharing agreement in compliance with the Ontario Personal Health Information Protection Act, the Research Ethics Board approved study aims and written informed consent signed by the participants. The Health Sciences Research Ethics Board of Western University should be contacted at ethics@uwo.ca for data-sharing related requests. The code is available at our Github repositories: https://github.com/IVarha/stn_segmentation and https://github.com/IVarha/MER_lib.

**Funding:** The study was supported by the grant of the Czech Ministry of Health, grant No. NV19-04-00233 (project CLIMABI), by the Brain Dynamics, grant number, CZ.02.01.01/00/22_008/0004643, Research Centre for Informatics, grant number CZ.02.1.01/0.0/16_019/0000765 and by the grant Biomedical data acquisition, processing and visualization, number SGS22/165/OHK3/3T/13., The funders had no role in study design, data collection and analysis, decision to publish, or preparation of the manuscript.

**Competing interests:** The authors have declared that no competing interests exist.

## Conclusion

The proposed method combines image-based segmentation of the subthalamic nucleus with microelectrode recordings to estimate their mutual location during the surgery in a fully automated process. Apart from its potential use in clinical targeting, the method can be used to map electrophysiological properties to specific parts of the basal ganglia structures and their vicinity.

## I. Introduction

The importance of treatment techniques in complex neurodegenerative disorders such as Parkinson's disease (PD) has increased with the ageing of the population. PD is characterised clinically by tremors, muscle rigidity and other motor symptoms resulting from the loss of dopaminergic motor neurons in Substantia nigra (SN), as well as by a range of other non-motor symptoms [1]. Currently, only symptomatic treatment is available to improve the quality of life of PD patients. Deep brain stimulation (DBS) surgery is used as a symptomatic treatment of motor symptoms in severe cases of late-stage PD. Accurate electrode positioning is vital for DBS surgery, directly influencing clinical outcomes.

Since the introduction of DBS for PD, the subthalamic nucleus (STN) has been established as one of the primary and most used surgical targets [2, 3]. Naturally, localisation of the target nucleus is required for surgical planning. This is typically based on presurgical T2-weighted magnetic resonance images (MRI), where the STN is characterised by hypointensity laterally to the red nucleus (RN). The border between the STN and the neighbouring SN is of low contrast, and the posterior boundary of the STN is very hard to distinguish in the T2-weighted image [4]. Segmentation of the STN and neighbouring structures is also challenging due to its small dimensions (less than 10mm along its longest axis, [5], making it coarsely delineated in typical clinical MRI images with slice thickness.

While indirect targeting techniques are also used for PD DBS, the manual identification of STN and SN locations from MRI images based on surrounding sttructures is a common way to identify the appropriate DBS target coordinates. Accurate bouvdary definition could assist in choosing of an optimal target point. However, this requires expert training for shape displacement definition and alignment with neighbouring structures. Manual labelling of clinical images is also problematic due to the relatively small size of the subthalamic nucleus (130mm3, on average) [6], which is especially crucial for clinical data.

Several recent papers have focused on MRI-based STN segmentation [7–10], but most of them are not based on data from clinical scanners. There are two possible approaches to the segmentation of deep subcortical structures: i) co-registration to the atlas [11] and ii) segmentation using image properties of these structures. Park [9], built a model for targeting a single T2 image slice, and they used a deep neural network approach to generate the STN as a pixel mask on the image and estimate the target from it.

Statistical Shape Models (SSM) became popular for subcortical segmentation, and some SSMs have been included in standard research packages, namely, the FMRIB Software Library (FSL) [10, 12]. These models use the Active Appearance Model (AAM) [13] to fit the 3D shape of subcortical structures on a newly-presented subject. AAM represents the intensity properties of the image as a statistical shape model. Patenaude [12] utilised a Bayesian framework to model intensity and shape relations. In the Patenaude model, the fitting procedure relies on

minimising the negative log-likelihood of a pre-trained shape conditioned on the intensity sampled from a new image.

Visser presented a statistical model which does not require manual labels for the training process [14]. The method requires only a reference mesh derived from an atlas or from a single label for initialisation. In addition, the model requires a user-defined intensity shape in each MRI modality around the anatomical boundary of each nucleus type. The flexibility of this method is suitable for finding unbiased differences between various groups of patients, which cannot be done on supervised methods trained on a single group.

DBS surgery is a complex and invasive procedure with several sources of bias ("brain shift"), which often cause the anatomical situation during the surgery to deviate from the MRI-based presurgical plan. One of the primary sources of this bias is caused mostly by cerebrospinal fluid (CSF) leakage, which may cause the STN to shift posteriorly up to 2 mm, leading to inaccurate targeting, especially during implantation of the second hemisphere [15]. In order to increase targeting accuracy, surgeries typically include intraoperative monitoring techniques to compensate for this bias. In most centres, this is achieved by electrophysiological exploration, which involves inserting multiple microelectrodes and recording the electrical activity in their vicinity (microelectrode recordings–MER) [16]. MER signals are reviewed during surgery by experts, primarily neurosurgeons or neurologists, who label structures by their characteristic signal patterns, such as increased firing rates and higher intensity of the neuronal background.

Many studies have focused on automatic MER-based nuclei classification in DBS [17–21], suggesting that automatic intra-operative MER analysis may improve the clinical outcome of the surgery and may reduce the surgery time, as neurosurgeons could spend less time on manual signal classification. Moreover, the surgeons do not need to be experts in the signal patterns of different structures [22]. Similarly to image-based STN segmentation, most models are based on supervised learning using manually labelled data, dependent on the manual inputs of an expert. The accuracy of expert labels thus limits the classification quality.

The challenge addressed in this paper is to combine the two different types of modalities, MRI and MER, in order to identify the most likely configuration of the STN with respect to the microelectrode recording sites intra-operatively. This approach allows us to directly estimate the extent and the direction of several sources of bias, especially brain shift. If implemented in clinical practice, automatic estimation of the MER location within the STN in 3D would be possible intra-operatively. Decision-making about the stimulation lead placement might be made more accurate, and valuable surgical time could be saved. The bottleneck restricting this area of research is the low availability of datasets combining both MRI data and intra-operative MERs.

In our previous research [23], we used the energy of the MER signal combined with a probabilistic model and maximum likelihood estimation to fit a 3D STN atlas to intra-operative electrophysiological MERs and provide automatic localisation of the MERs within the STN. In this case, an STN atlas was modified using 9 degrees of freedom, including translation rotation and scaling. We modified this method for the semi-automated segmentation model using Active Contours [24].

In this study, we implement a fully automatic segmentation algorithm, which combines MRI-based segmentation of the structures of interest (i.e. the STN, the SN and the red nucleus —RN), followed by refinement according to intra-operative MER data.

## II. Methods

The model presented in this work is divided into two steps. First, automatic 3D segmentation of three basal ganglia structures: the STN, SN and RN, is performed on the patient's pre-

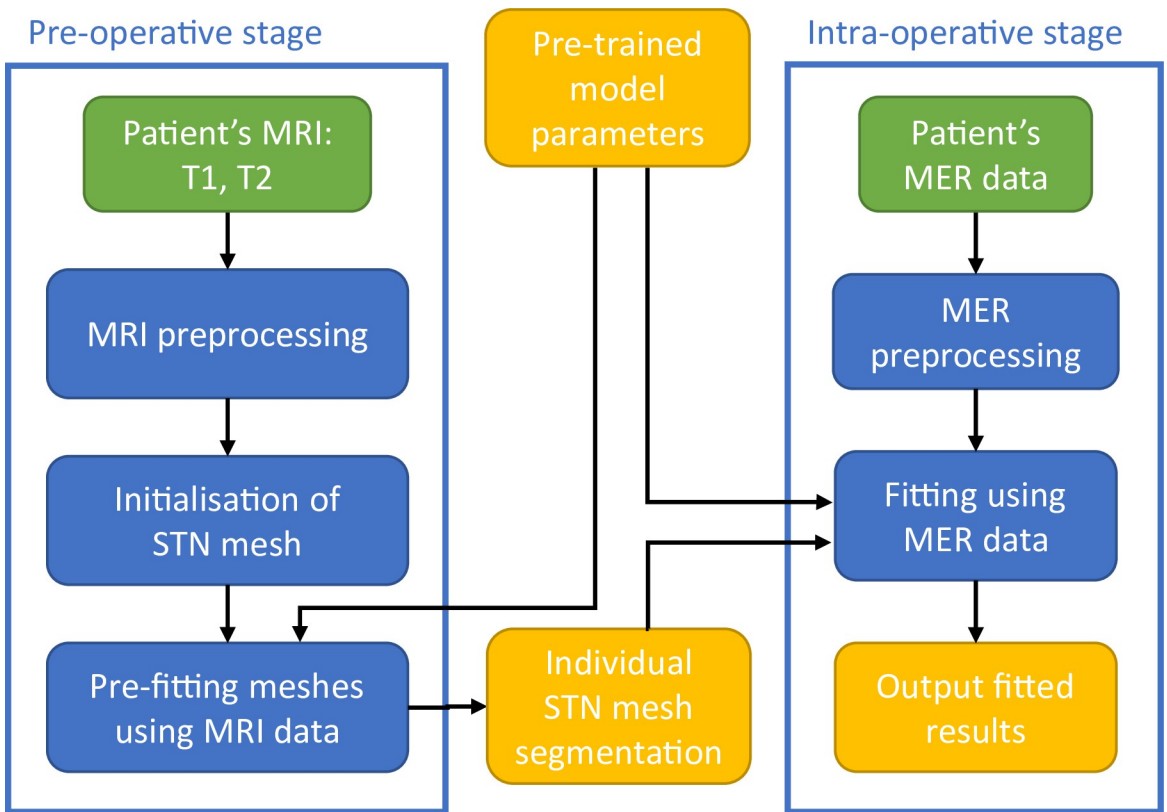

**Fig 1. Schematic overview of the operating principle of the pre-trained model when applied to a novel patient's data.** Overview of model pre-training, as well as MRI and MER preprocessing are detailed in Fig 2.

operative MRI data (image-based model). Second, this individual 3D segmentation is used to automatically identify the location of exploratory electrodes with respect to these structures, using electrophysiological properties given by the intra-operative MERs (electrophysiology-based model). Prior to model evaluation, the parameters of the model are estimated on training data with annotated contours of the selected basal ganglia structures and MER signals labelled as inside or outside the STN. Then the model is evaluated on unlabelled data to provide automatic segmentation of the basal ganglia structures and localisation of the microelectrodes with respect to the STN. An overview of the evaluation-stage steps can be found in Fig 1.

## A. Data

To develop and evaluate the model, we used a dataset of 120 PD patients, 38 Females and 82 Males, who underwent bilateral DBS surgery at the University of Western Ontario, Canada. The age of the patients ranged from 38 to 75, with an average age of 60.52±6.76, and disease duration from 4 to 28 years, with an average of 11.13±3.98. Electrophysiology MER data was available for a subset of 45 patients. The MER patients are 16 females and 29 males, aged between 38 and 75 with an average of 59.87±6.59, and disease tracking from 4 to 22 years with an average of 10.29±4.15. The study was approved by the Human Subject Research Ethics Board office at the University of Western Ontario (ethics #109045) and performed according to all relevant regulations. All participants signed an informed written consent including agreement to use their clinical data for research purposes. The study included male and female PD patients between 38–75 years of age on stable doses of anti-Parkinson medication.

Exclusion criteria were the history of surgical intervention for PD (i.e., previous DBS or lesion) and presence of dementia or any other condition that prevents the ability of the participant to provide fully informed consent. The data were collected between 07/2008 and 05/2019 and accessed for current research between 09/2022 and 02/2023. Greydon Gilmore managed the data, all other authors had access only to anonymised data.

For each patient, axial T2-weighted (T2w) MRI images (echo time = 110 ms, repetition time = 2800 ms, receiver bandwidth = 20.83 kHz, field of view = 26 cm, matrix size = 256 × 224, slice thickness = 1.5 mm, resolution = 1.25 × 1.25 × 1.50 mm) and post gadolinium-enhanced volumetric T1-weighted (T1w) images (echo time = 1.5 ms, inversion time = 300 ms, flip angle = 20˚, receiver bandwidth = 22.73 kHz, field of view = 26 cm x 26 cm, matrix size = 256 × 256, slice thickness = 1.4 mm, resolution = 1.25 × 1.25 × 1.50 mm) were obtained 2 weeks prior to surgery (Signa, 1.5 T, General Electric, Milwaukee, Wisconsin, USA). Postoperative computed tomography (CT) (tube voltage = 20 kV, tube current = 145 mA, data acquisition diameter = 1,331 mm, reconstruction diameter = 320 mm, matrix size = 512×512 voxels, pixel spacing = 0.625 × 0.625 mm2, axial slices = 96, slice thickness = 1.25 mm, gantry tilt = 0o; LightSpeed VCT, GE Medical Systems, Chicago, IL, USA) was acquired once the DBS leads were in place. A surgical plan, including the entry and target point and the configuration of the Leksell stereotactic frame, was collected for each patient. A preoperative CT image with a stereotactic frame was available to identify the stereotactic coordinate system.

During intra-operative electrophysiological exploration, a computer-controlled microelectrode drive was mounted to the stereotactic frame (StarDrive, FHC Inc., Bowdoinham, ME), and 2–5 cannulas with tungsten microelectrodes (60μm diameter) were lowered to 10.0mm above the surgically planned target. Electrophysiological signals (MER) were recorded in increments of 1.00mm (10.00mm to 5.00mm above the surgical target) and 0.50 mm (5.00 mm above the target until the SN was reached, marking the ventral STN border). This resulted in approximately 25–30 recordings for each microelectrode. Data were collected for 10 seconds at each recording site. The signals were sampled (24 kHz, 8-bit), amplified (gain: 10000) and digitally filtered (bandpass: 500–5000 Hz, notch: 60 Hz) using the Leadpoint recording station (Leadpoint 5, Medtronic). Neurosurgeons and neurologists labelled recording sites within the STN during surgery based on the MER signal, and the most appropriate trajectory and stimulation contact depth were chosen and recorded.

## B. MRI data preprocessing

**1) Data co-registration.** All MRI data was first coregistered into the same space using the MNI152NLin2009bAsym template with a resolution of 0.5x0.5x0.5mm [25]. For better alignment in the subcortical area, we used a two-stage linear coregistration process described by Patenaude [12]: Initial pre-alignment is performed in the first stage, followed by the next stage of coregistration with subcortical mask weighting. Both procedures were carried out with 12 degrees of freedom (DOF) and were performed using FLIRT from the FSL package [26]. The T2 images were first coregistered to T1 on an individual level, and then the transformation to MNI152 space based on T1 images was applied to them.

The linear procedure was chosen due to natural affine transformations, which preserve reversible point correspondence. Moreover, with linear coregistration, we could operate with non-interpolated intensities in the patients' native space.

**2) Intensity normalisation.** To model the statistical image-based properties of the basal ganglia structures, we used intensity normalisation to suppress intensity inhomogeneities between subjects. Before intensity normalisation, we used a robust brain extraction tool (BET) from the FSL package [27] to exclude image regions other than the brain itself.

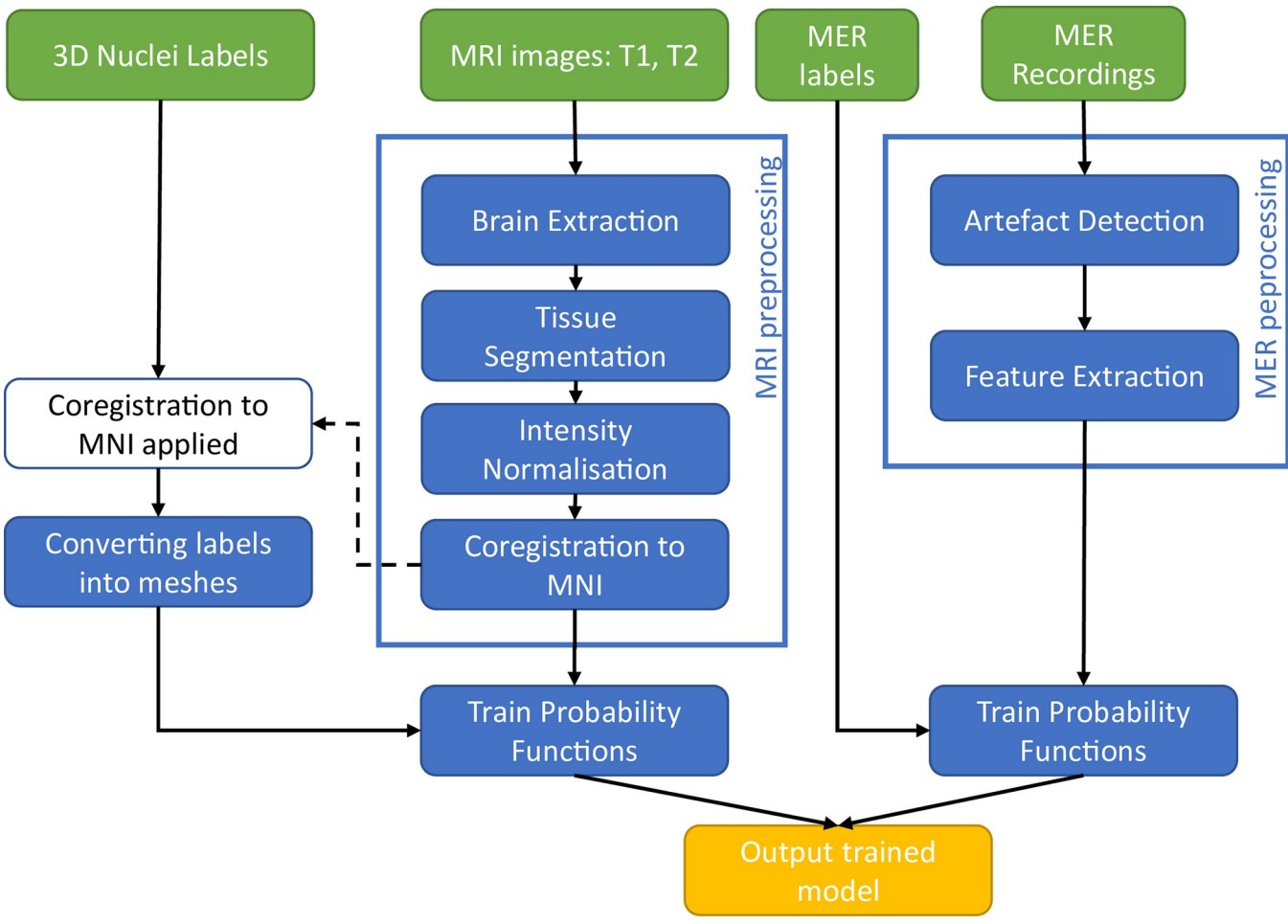

**Fig 2. Pipeline for the pre-training phase of the model, including MRI and MER data processing.**

All three basal ganglia structures used in this study are distinguishable in the T2 weighted images and are located in the thalamic area surrounded by white matter. We used fuzzy c means over the white matter to normalise the T2 image intensity [28]. This is advantageous, as it allows us to focus normalisation on the white matter only. Besides the image itself, the normalisation procedure requires a white matter mask, which contains binary values that determine whether a voxel belongs to white matter. We generated the masks using the FMRIB's Automated Segmentation Tool (FAST) [29]. This normalisation procedure helps to maintain comparable intensity values in the basal ganglia structures of interest across different participants.

The whole data preprocessing and model fitting process for both modalities is summarised in Fig 2.

**3) Manual labels.** Manual labels of the basal ganglia structures were derived as voxel-level regions of interest (ROIs) by a trained expert using ITK-snap software (1.8.1), based on coregistered T1-weighted volumetric images and T2-weighted slab images. The low signal-to-noise ratio in the clinical MRI data affected the visibility and the contrast of the structures. To overcome this, we assessed our manual labels with a blind relabelling procedure for a subset of 15 subjects. The two sets of labels were compared using the dice score (below). In this way we estimated the labelling error, representing the upper bound of automatic classification accuracy.

Throughout this text, to compare the overlap between two meshes, we compared the encapsulated volumes using the dice score:

$$Dice = \frac{2*card(X\ or\ Y)}{card(X) + card(Y)}, \tag{1}$$

where X and Y are binary masks of the label, card–is the cardinal function that calculates the number of elements in the set of data, which is a number of positive voxels in a mask.

**4) Generation of 3D meshes from labels.** The proposed algorithm works with a 3D mesh —a structure defining a 3D surface using a set of vertex point coordinates (vertices) and surface polygons defined as sets of connected vertices (surfaces). The voxel-level nuclei labels were thus transformed to surface meshes using a method proposed by Patenaude [12], which guarantees that the meshes across subjects maintain an approximate point correspondence. In our data representation, vertex coordinates are concatenated with intensity profiles to represent the statistical shape model for which the point correspondence is needed.

The inputs to the mesh generation process are MRI volume labels presented as 3D ROI sets of voxels, where the voxel value represents the voxel membership either of the given structures (ones) or of the background (zeros). As our images were registered to MNI152NLin2009bAsym during the previous processing stages, processing of the labels is carried out in this space. For each anatomical structure (a single ROI), we first calculate the centre of mass (C = [cx, cy, cz]) of our label, defined in each coordinate (x,y,z) as:

$$Cx = \frac{\sum_{i=1}^{Nx} i*(\sum_{j,k=1}^{Ny,Nz} V(i,j,k))}{\sum_{i,j,k}^{Nx,Ny,Nz} V(i,j,k)}, \tag{2A}$$

$$Cy = \frac{\sum_{j=1}^{Ny} j*(\sum_{i,k=1}^{Nx,Nz} V(i,j,k))}{\sum_{i,j,k}^{Nx,Ny,Nz} V(i,j,k)}, \tag{2B}$$

$$Cz = \frac{\sum_{k=1}^{Nz} k*(\sum_{j,i=1}^{Ny,Nx} V(i,j,k))}{\sum_{i,j,k}^{Nx,Ny,Nz} V(i,j,k)}, \tag{2C}$$

where Nx, Ny, Nz is the number of voxels in each dimension of the binary image label V. Further, we initialise a spherical mesh centred in the centre of mass C with a radius larger than the maximum size of the given anatomical structure across the dataset. In the next step, we move each vertex towards the centre of the sphere until the interpolated label value at the vertex location reaches the threshold of 0.5. Once the sphere has shrunk to the boundaries of the voxel-based ROI for a given structure, Laplacian smoothing is applied, making the vertex distribution more even by using the average coordinate of all adjacent vertices–see Fig 3. The resulting mesh is subsequently transformed to the patient native space.

## C. Image-based model

**1) Basis of the statistical shape model.** After obtaining the mesh representation, it is possible to build a statistical model over the shapes and intensities using statistical modelling [12, 13, 30]. Considering that the vertex generation on the sphere is consistent, we expect the same vertices to have a very close anatomical position across subjects. To construct the model, we concatenate the vertex coordinates, e.g. for a line defined by two points l = {(0,1,2),(3,7,5)} the resulting representation of this line would be x = [0, 1, 2, 3, 7, 5].

For a training set $Z = \{x_1, x_2...x_n\}$ with $n$ subjects, we represent the vertex locations of each subject using Multivariate Normal Distribution, defined as:

$$p(x_i|\mu,\Lambda) = N_r(x_i|\mu,\Lambda) = (2\pi)^{\frac{-r}{2}} det(\Lambda)^{\frac{1}{2}} exp\left(\frac{-1}{2}(x_i - \mu)^t \Lambda (x_i - \mu)\right), \qquad (3)$$

where $r$ is sample dimensionality, $\Lambda$ is a precision matrix with dimensions $r$ x $r$ (pseudoinverse of the covariance matrix), and $\mu$ is the mean vector. Subsequently, we can estimate the parameters from a training dataset using the minimum covariance determinant (MCD) [31].

**2) Joint shape-intensity model.** Prior to this point, we have discussed shape representation only. Intensity representation in our model assumes that boundaries share a similar intensity pattern across subjects. Moreover, these patterns may vary for different parts of the segmented structure.

Using cubic spline interpolation from the voxel space, we calculate the intensity profiles within 3.0mm along the vertex normal in both directions (inside and outside the STN) at a total of 7 evenly-spaced locations. Then the measured intensity values are normalised by subtracting the average intensity of the structure in a given patient.

Next, we concatenate the intensity features with the shape, as explained in the previous section, so our feature vector will be $x_i = [x_{shp}, x_{int}]$. Assuming independence between intensity and shape from (2), we obtain that probability density function, presented as:

$$p(x_i|\mu,\Lambda) = N_r(x_i|\mu,\Lambda) = N_r(x_{shp}, x_{int}|\mu,\Lambda) = N_{r_{shp}}(x_{shp}|\mu_{shp},\Lambda_{shp}) \cdot N_{r_{int}}(x_{int}|\mu_{int},\Lambda_{int}) \quad (4)$$

Where $r_{shp}$, $r_{int}$−dimensionality of the features vector for shape and for intensity, respectively.

**3) Feature space decomposition.** For shape representation, it is natural that the location of one vertex is dependent on the location of other adjacent vertices. Hence, utilising these dependencies in the data can significantly reduce the number of variables in the feature vector. In order to reduce the dimensionality of the dataset, we used Singular Value Decomposition, allowing us to pick a limited number of principal eigenvectors [32], thus effectively reducing the size of the dataset. We form a matrix M = [x1-μ, x2-μ...xn-μ] by concatenating demeaned samples from our training dataset Z. Following that, we decompose matrix M using SVD and obtain a matrix of eigenvectors V = [v₁,v₂...vₙ] and a matrix of eigenvalues (on diagonal) matrix S. As a result, we obtain the decomposition of a sample $x_i$ as follows:

$$x_i \approx \overrightarrow{\mu} + \sum_{j=1}^{L} a_{ij} * \overrightarrow{v}_j \qquad (5)$$

where $L$ is the chosen number of most important eigenvectors, representing most of the variance, and $v_j$ is an eigenvector. In our case, we chose the number $L$ to explain more than 99.5% of the variance in the dataset, allowing us to reduce the calculation time and the number of model parameters significantly while at the same time preserving most of the variability. The resulting vector of the decomposition $A = \{a_i\}$ forms a new reduced feature vector.

**4) Fitting procedure.** To perform a fit on a new subject, i.e. to segment the given nuclei from the newly-presented unlabelled data, we formulate the following procedure, based on minimisation of the Mahalanobis distance between the presented data and the distribution estimated previously in the training process. By calculating the Mahalanobis distance for Eq

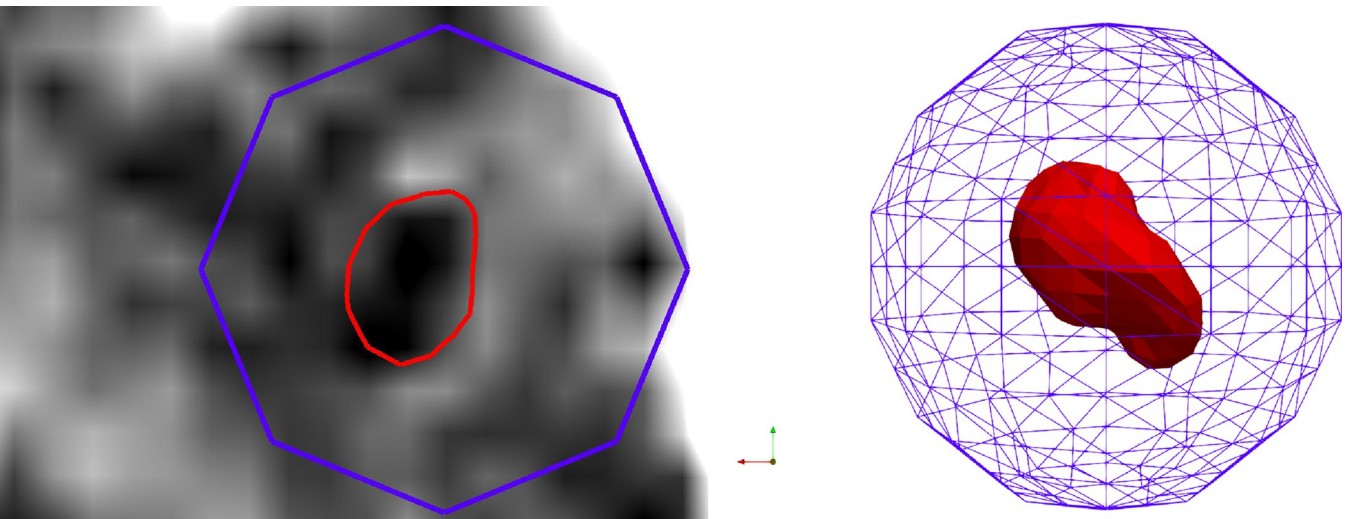

**Fig 3. Representation of the shrinkage process in a patient native space.** The purple mesh is an ellipsoid that resulted from sphere initialisation on the MNI coordinates. A red mesh–resulting mesh representation of a 3D STN label.

(3), we obtain:

$$Mah^2(x_i|N_r(\mu,\Lambda)) = Mah^2|x_{shp})N_{r_{shp}}(\mu_{shp},\Lambda_{shp})) + Mah^2(x_{int}|N_{r_{int}}(\mu_{int},\Lambda_{int}))$$

$$= (x_{shp} - \mu_{shp})^T\Lambda_{shp}(x_{shp} - \mu_{shp}) + (x_{int} - \mu_{int})^T\Lambda_{int}(x_{int} - \mu_{int}) \qquad (6)$$

Eq 6 employs both the distribution for shape and the distribution for intensity. Moreover, if we were to add other independent modalities (such as electrophysiology) or other constraints, we could do so by adding more Mahalanobis distance components to Eq (5). A possible constraint is the mean intensity of the structure. When we use parametrisation with eigenvectors, a typical restriction would be on features coordinates. To preserve most data, suitable limits would be $-3\sqrt{\lambda_j} \leq b_j \leq 3\sqrt{\lambda_j}$, where $\lambda_j$–is the eigenvalue corresponding to the variance of feature $j$. When we work with meshes, it is natural to prevent self-intersections. By using the eigenvectors to represent vertex coordinates, intersections are effectively prevented.

The crucial part of the fitting procedure is appropriate initialisation. A possible way is to initialise the starting mesh at a mean vector or a median sample. We used multiple initialisations to avoid convergence to a local minimum. Four of our initialisations are on $\pm\lambda$ of the two most essential eigenvalue subjects, and one is the median value.

To perform the fitting procedure, we used the Powell method [33]. This method does not require the numerical calculation of gradient values, but this is at the expense of a higher number of iterations. The imaging data is retrieved before DBS surgery, so this data can be processed before the surgery, the initial segmentation can be produced without microelectrode data.

The model was implemented as a Python module with C++ embedded source code.

**5) Distributions of joint structures.** The framework presented here can work with multiple structures simultaneously: by doing this, we can utilise the mutual location between structures to achieve better fitting results, especially for structures that are located close to each other. The processing does not differ from that in a single nucleus, but we estimate all values independently for each structure, using concatenated vectors.

To process multiple structures in subject data, we can utilise the Bayesian probability rule, where the shape of a single structure is correlated with others. For shapes $x_1$ with already present $x_2$, and considering that both shapes are distributed using the multivariate normal distribution, we obtain that $x_1$ is also distributed by a normal distribution described by:

$$p(x_1|x_2) = N_r(x_1|\mu, \Lambda\Lambda)$$

$$\mu = \mu_1 - \Lambda_{11}^{-1}\Lambda_{12}(x_2 - \mu_2)$$

$$\Lambda = \Lambda_{11}$$

Where $\Lambda_{11}$ is a precision matrix of shape one and $\Lambda_{12}$–is a precision matrix that describes the correlation between shapes 1 and 2. In the present study, we used joint distribution for the STN and SN structures to increase the segmentation accuracy despite the indistinct boundary between these structures.

**6) Surface based statistics.** The approach presented above allows an analysis of the intensity characteristics around various structures of interest. We performed k-means clustering of intensity profiles around STN surfaces derived from the manually labelled data to visualise and localise characteristic intensity profiles on the STN. The mean intensity profiles of each cluster, together with the cluster membership of each STN surface vertex, were visualised to provide an overview of the surface intensity characteristics.

To provide another example of the utility of our model, we built a likelihood map of STN stimulation electrode entry and exit on the STN surface. For this purpose, we used the stimulation electrode artefact in the coregistered postoperative CT images, and we calculated the proportion of cases when the electrode crossed a particular triangle on the STN surface. Further we mapped a points of entry and exit to a standard space. As the procedure of mesh generation maintains point correspondence, the points are mapped using their location within the face triangle to same numbered face on average STN. Futher, we fitted a 3D multivariate normal distribution based on the mapped intersection count to average STN mesh points and computed density function value to the center of faces for visualisation puposes.

## D. Electrophysiology-based model

**1) MER based brain-shift correction model principles.** MER is treated as an additional independent modality, which is not used during the image segmentation of the STN shape but is employed in a subsequent step to estimate the brain shift and determine the intra-operative microelectrode locations with respect to the segmented surfaces during surgery. The MER part of the proposed method is initialised with the STN surface segmented from the MRI data and the micro electrode recording sites arranged according to the planned target and stereotactic frame settings. We assume that the much higher spatial resolution of MRI data provides a reasonable estimate of the STN shape, and we use the MER part of the model to translate the individually segmented STN only. Additionally, scaling of the segmented STN shape according to MER fit is allowed separately for individual dimensions (i.e. separately along the x, y and z axes), in order to allow for possible mismatch between the dimension of MRI hypointensity and electrophysiological STN. Also, the separate scaling in individual direction allows to compensate for possible labelling and segmentation inaccuracies caused by anisotropic resolution of the MRI data.

The evaluation of the MER model was performed separately using the leave-one-subject-out scenario and the performance was calculated according to the agreement with expert labels of the individual MER recordings.

**2) MER preprocessing.** The MER preprocessing started with the digitised and bandpass-filtered RAW MER recordings described above. To extract artefact-free electrophysiological activity, stationary segments of the electrophysiology data were first identified using the covariance method [34], and the root-mean-square value was calculated from the stationary segments for each signal. Next, the normalised root mean square (NRMS) was calculated for each electrode trajectory by dividing the RMS values at all depths by the mean RMS from the first five recording positions as a reference [19]. In this way, the effect of varying electrode impedance or non-uniform amplifier gain across the dataset is minimised. In this paper, all preprocessing of MER data was done retrospectively after the surgery. However, the same procedures can be applied in real-time for feature extraction. All MER processing and fitting were performed using Matlab 2021a (MathWorks Inc., Natick, MA, USA).

**3) The MER model structure.** The MER model uses a probabilistic framework based on NRMS values [23, 35], which assumes two states: inside the STN and outside the STN, with different distributions of NRMS values.

We call the distribution of the NRMS values in each of the states the "emission probabilities" and represent them parametrically using the log-normal distribution (Fig 4 RIGHT). The smooth transition at STN entry and exit is modelled using a sigmoid function of the distance to the STN entry/exit (Fig 4 LEFT) $S(d_i|\Theta) = (1 + \exp - (\beta^0 + \beta^1(d_i)))^{-1}$, where $di$ is the Euclidean distance to the nearest point on the surface of the STN mesh model at its currently assumed position, and $\Theta$ are the model parameters. The likelihood of each observation is then a function of the recorded NRMS value $xi$ at a distance to the assumed STN surface $di$, and is given by:

$$p(\{x_i, d_i|\Theta) = p(x_i|IN) \cdot S(d_i|\Theta) + p(x_i|OUT) \cdot (1 - S(d_i|\Theta))$$

where $p(x_i|IN)$ and $p(x_i|OUT)$ are the likelihoods computed using the log-likelihood probability density function using the trained parameters for inside and outside the STN, and $S(d|\Theta)$ is the sigmoid function for the distance to STN–$d$, given pre-trained parameters $\Theta$.

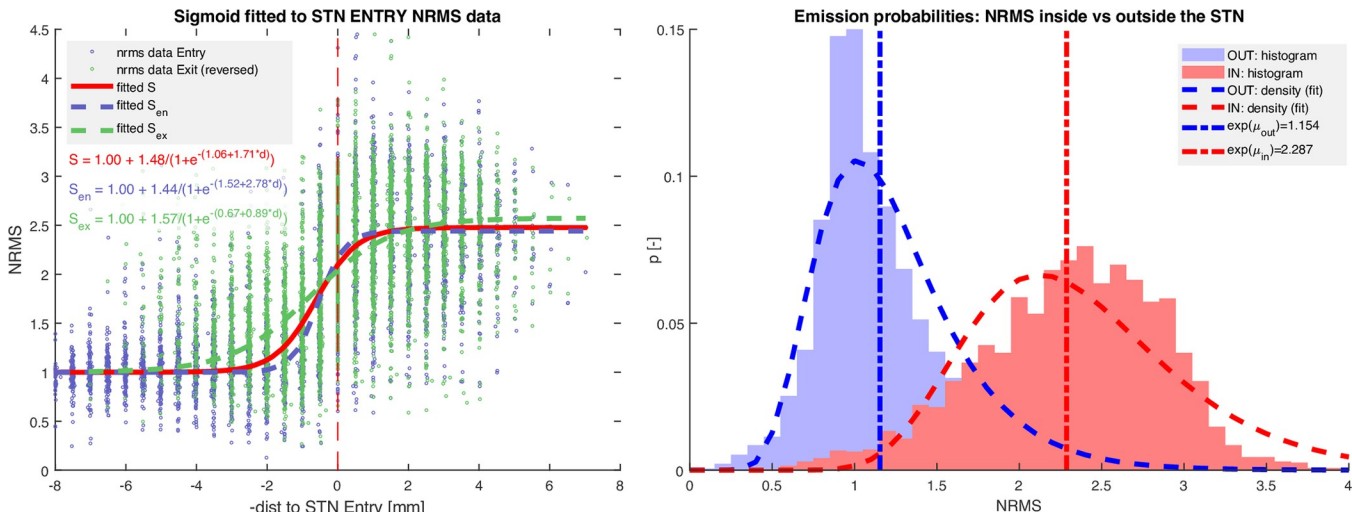

**Fig 4.** *LEFT*: Sigmoid transition function (red), modelling the smooth increase in signal energy (NRMS) around the STN border (red dashed vertical line at 0 mm) overlaid on recorded values around the STN entry (blue dots) and mirrored data around the STN exit (green dots). It is apparent that the fitting of separate sigmoids to the STN entry (dashed blue) and the STN exit (dashed green) data separately results in a considerably steeper rise at the STN entry. However, for simplicity, we use only the mean trend (red line) for all points on the STN surface. *RIGHT*: Probability distribution of the normalised signal energy (NRMS) in the two states: inside the STN (red) and outside the STN (blue), each modelled using a log-normal distribution.

Next, a maximum likelihood electrode position with respect to the segmented structure is sought using translations and scaling of the electrode locations in order to minimise the negative log-likelihood of the observed NRMS data, as follows:

$$t^* = argmin \sum_{i=1}^{L} -ln(p(\{x_i, l_i\}|t, \Theta)) \tag{6}$$

where t* is the resulting translation and scaling vector along the x, y, z axes, $x_i$ are the L NRMS values measured at locations $l_i$ (i.e. the MER recording sites), and $\Theta$ are parameters of the probabilistic model estimated on training data. The resulting translation t* is then applied to the STN model and evaluated. Contrary to our previous work [23, 35], no rotation was done at this stage. Also, to avoid extreme deviation from the initial position, we used a constraint optimisation procedure to minimise the negative likelihood in Eq 6. The constraints were set to translation of max. +-2mm per axis and scaling between 0.9 and 1.2 per axis.

If we represent the MER model part as an additional modality in the intensity-based model from (5) and (6), and considering the independence of shapes intensities from MER data, we obtain:

$$
\begin{aligned}
Mah^2&(x_i|P(\sim)) \\
&= Mah^2(x_{shp}|N_{r_{shp}}(\mu_{shp}, \Lambda_{shp})) + Mah^2(x_{int}|N_{r_{int}}(\mu_{int}, \Lambda_{int})) \\
&\quad + Mah^2(x_{mer}|ln(\mu_{MER}, \sigma^2)) \\
&= (x_{shp} - \mu_{shp})^T \Lambda_{shp}(x_{shp} - \mu_{shp}) + (x_{int} - \mu_{int})^T \Lambda_{int}(x_{int} - \mu_{int}) \\
&\quad + (x_{mer} - \mu_{mer})^T \Lambda_{mer}(x_{mer} - \mu_{mer}),
\end{aligned} \tag{7}
$$

where $x_{mer}, \mu_{mer}, \Lambda_{mer}$ are observation, mean and precision matrices of the microelectrode recording distribution.

## III. Results

In the following section, we present the results of the Statistical Segmentation Model on clinical data from a group of PD patients. We separately evaluated the segmentation of anatomical structures and electrophysiology-based shifting, both using the leave-one-subject-out cross-validation (LOSO) strategy.

### A. Manual labelling quality

First, we assessed the quality of the manual labelling of the 3D contours of the anatomical structures by a manual relabeling procedure on a reduced dataset comprising 15 subjects randomly selected from the dataset. The resulting overlap for the STN was dice score 0.71±0.09, with the lowest value of 0.53.

### B. Intensity-based segmentation

First, we evaluated the overlap between the automatically determined structure boundaries and the manual labels for basal ganglia structures—see Fig 5. Automatic segmentation of the RN showed the highest accuracy among all structures, as RN is easily distinguishable on T2 images, while the STN turned out to be the most difficult to segment due to the lower contrast and the often unclear boundary with the adjacent SN. For this reason, we also used the above-mentioned joint distribution for STN and SN, while RN was segmented separately. An example of a segmented STN and RN contours overlaid on a clinical T2 MRI slice can be seen in Fig 6.

## Dice Scores Using Leave–One–Out Cross Validation

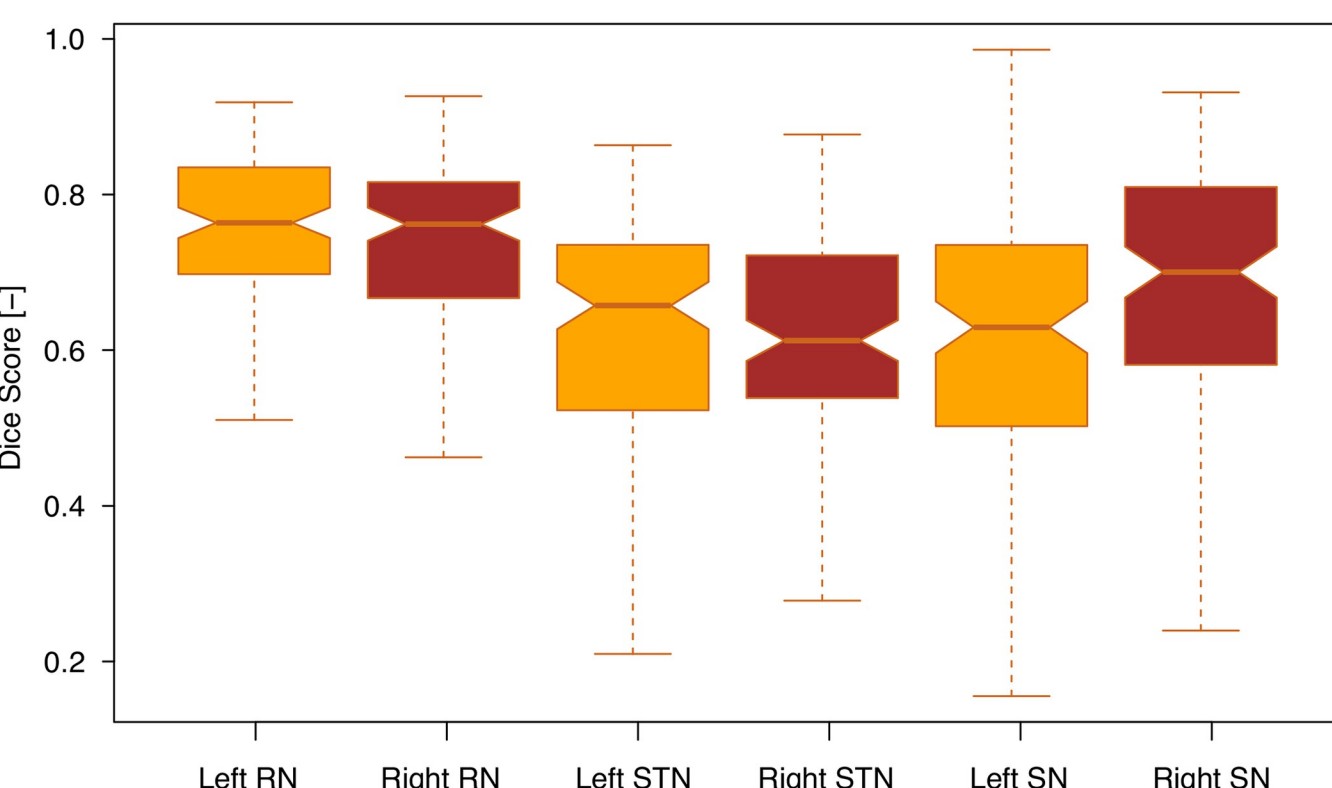

**Fig 5. Boxplots of the dice overlap between the segmented structures, namely, Subthalamic nucleus (STN), Substantia Nigra (SN), Red Nucleus (RN) and the manual labels of 120 subjects.** All structures were segmented using the statistical segmentation method presented here. For segmentation of STN and SN we merged their components as the boundary between them is barely visible.

All methods show the best overlap in the red nucleus in terms of dice scores. For STN, however, the scores are lower by 0.1. As can be seen in the first and third quartile in Fig 5, the spread of dice scores in our results is relatively high (lowest in the RN). This is likely due to the low contrast and resolution of the clinical 1.5T scans used in work presented here.

Most of the existing papers report segmentation results on higher quality data than those used in our paper. The presented clinical low field low-resolution data provides inferior results because the lower field strength provides a significantly lower signal-to-noise ratio. However, the results represent the accuracy achievable in clinical practice without changes to the current processes and equipment upgrades.

### C. Post-surgery statistical analysis

The point correspondence, which our method maintains, allows us to compute regional statistics on the STN surface–such as typical intensity profiles in various STN regions or typical entry and exit points of the stimulation electrode. To identify characteristic MRI intensity profiles along the STN border, we used the clustering of mesh vertices across our dataset. This provided the most consistent results when clustering into four clusters. Marked differences can be observed across different sub-regions of STN (see Fig 7).

Next, we evaluated the distribution of the stimulation electrode entry and exit along the STN surface (Fig 8). The evaluation procedure for the distribution of the entry and exit was

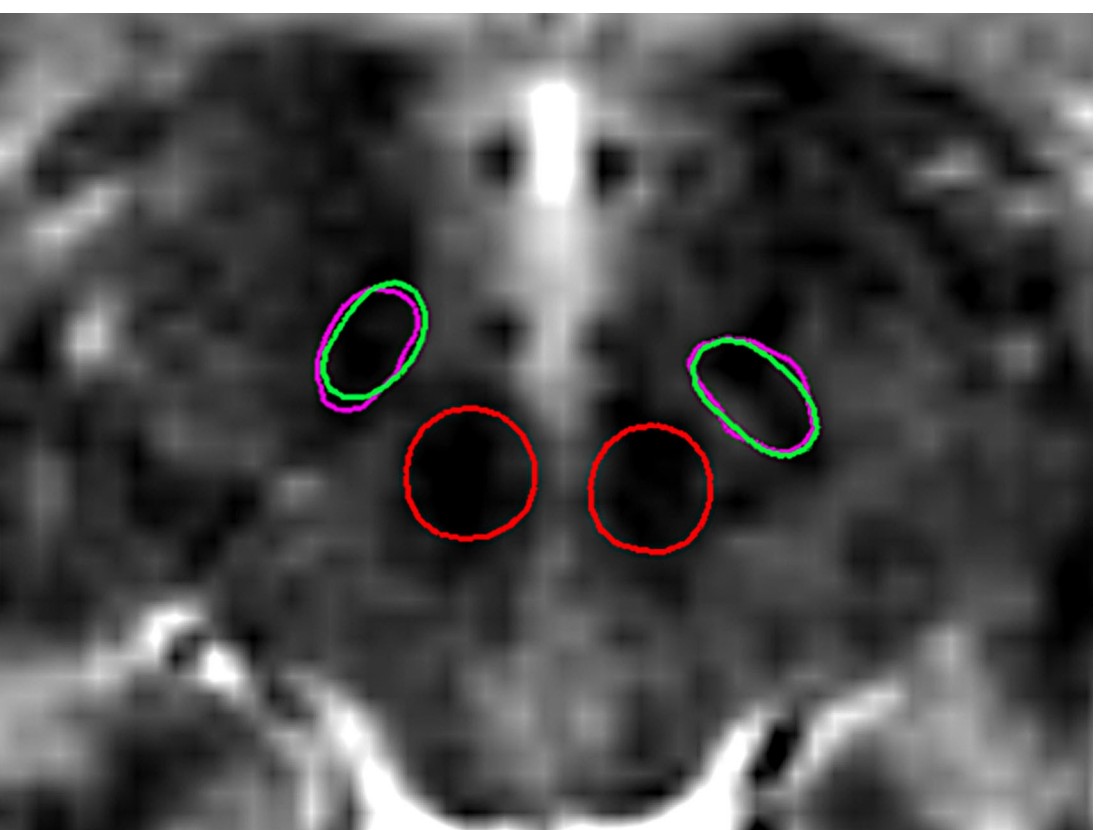

**Fig 6. Segmentation result for STN (green) in comparison with manual label (purple) overlaid on an axial T2-weighed MRI slice.** For illustration purposes Red Nucleus is also included (red). The contours represent cross-sections of the 3D mesh labels.

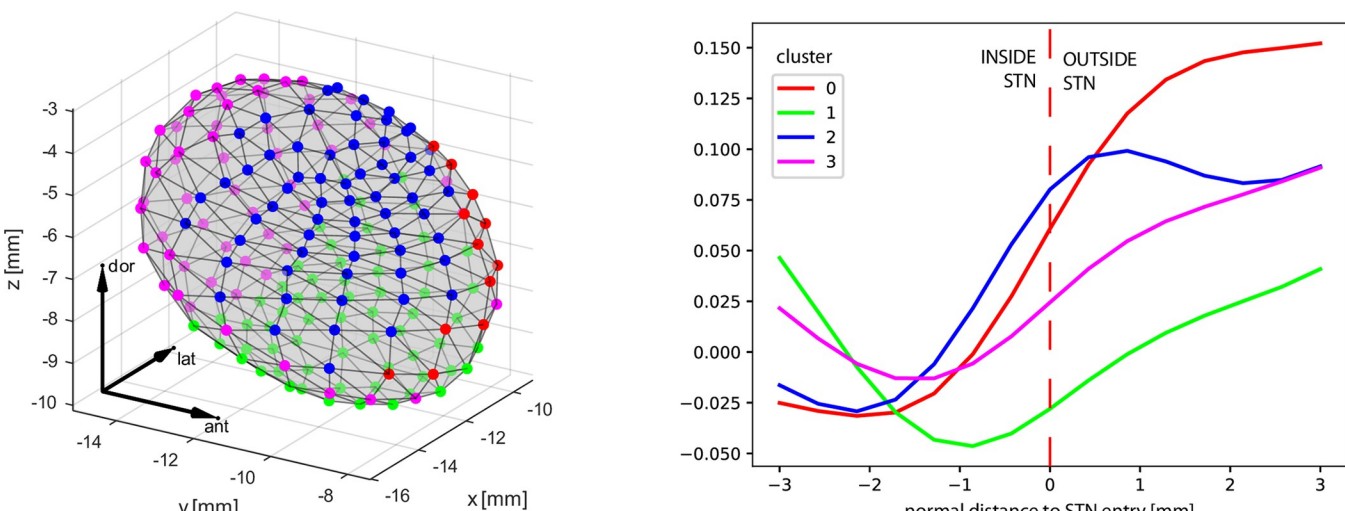

**Fig 7.** Clustering of T2 MRI intensity profiles around the STN border into 4 clusters: their localisation on the mean right STN surface (LEFT) and the corresponding intensity profiles–mean per cluster (RIGHT) 3mm along the vertex norm. Steeper changes in image intensity around the STN border are apparent in the anterodorsal part of the STN (red and blue clusters). The mean STN shape is shifted arbitrarily, coordinates in milimeters.

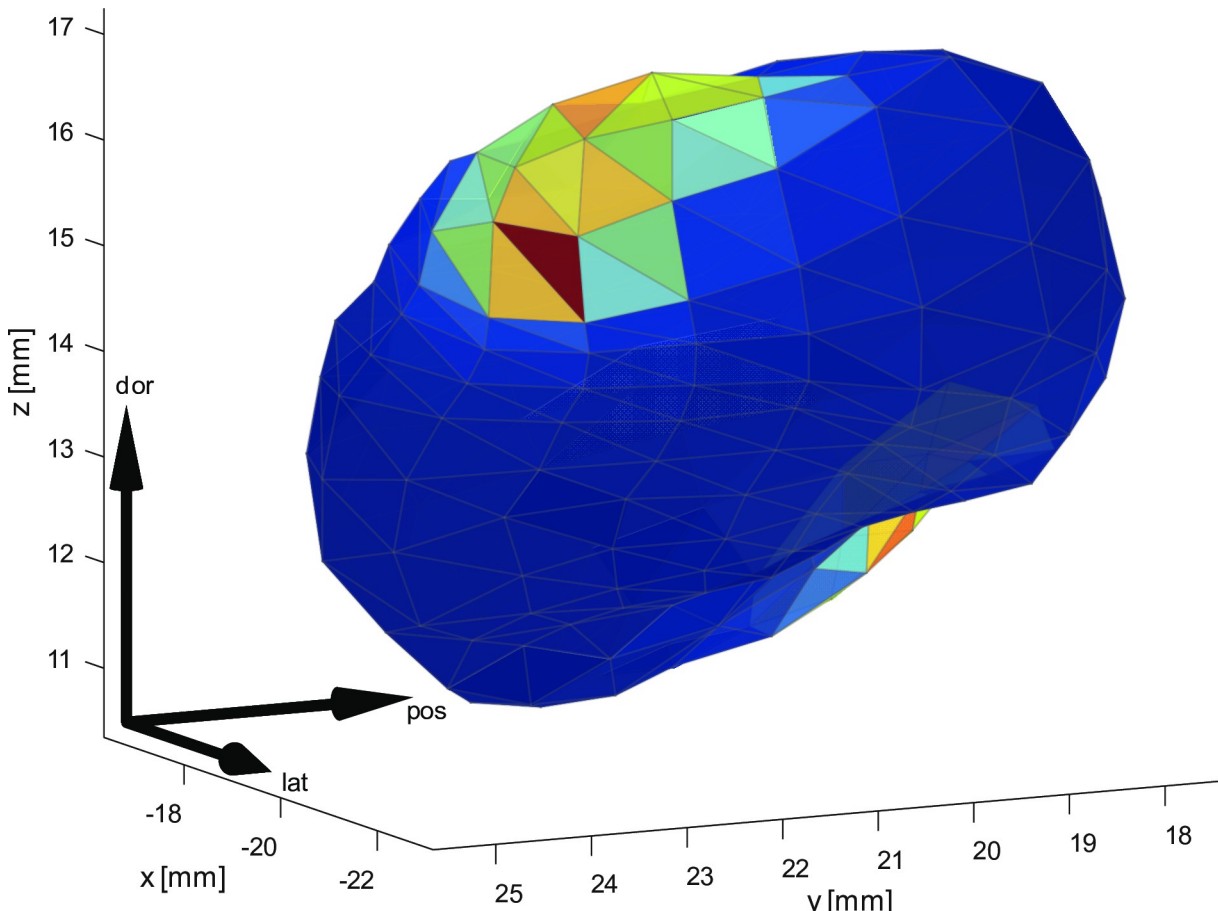

**Fig 8. Jet plot of likelihood of permanent electrode entry and exit, visualized on a STN mesh of an individual patient.** The calculation was based on trajectories determined using the electrode artefact in postoperative CT images. After coregistration to patient native MRI space, intersection with individual segmented STN 3D mesh was calculated for each patient. Owing to the point correspondence maintained by the proposed model, normal distribution of the electrode entry and exit points was calculated on the STN surface and transferred back to individual patient's native space.

computed by identifying the 3D mesh triangles intersecting with the final microelectrode trajectory entry and exit point in patient native space as described in section II.C.6) Surface Based Statistics. When combined with the model presented here, this distribution can be mapped to the individually segmented STN used for trajectory selection during the surgery. We also further evaluated the suitability of this surface distribution for initialising the electrode trajectories in the MER-based fit (i.e. the max likely exit initialisation method).

## D. Full model with mer fitting

Once the structures had been segmented using the MRI-based part of the model, we performed electrophysiology-based shifting of the assumed location of the exploration microelectrodes. This approach simulates the intra-operative situation when the individual STN has been segmented from the pre-operative MRI data, and the team now works to determine the optimal stimulation site using intraoperative microrecording.

The results presented in Fig 8 and Table 1 present the overlap between the surgeon-determined location of a given recording site (STN or other) and containment of that site in the

**Table 1. Test-set results of MER-MRI fitting for manually and automatically segmented 3D contours, using different initialisation methods and with/without MER-based shift.**

| Segmentation | Initialisation | MER-fit | Accuracy | | Sensitivity | | Specificity | | Youden J | |
|---|---|---|---|---|---|---|---|---|---|---|
| | | | mean | sd | mean | sd | mean | sd | mean | sd |
| manual | surgical plan | No | 0.778 | 0.075 | 0.438 | 0.231 | 0.889 | 0.065 | 0.327 | 0.240 |
| | max likely exit | No | 0.785 | 0.071 | 0.495 | 0.190 | 0.877 | 0.059 | 0.372 | 0.215 |
| | surgical plan | Yes | 0.844 | 0.062 | 0.532 | 0.234 | 0.932 | 0.071 | 0.464 | 0.221 |
| | max likely exit | Yes | 0.850 | 0.054 | 0.549 | 0.227 | 0.936 | 0.062 | 0.486 | 0.213 |
| automatic | surgical plan | No | 0.770 | 0.075 | 0.419 | 0.224 | 0.878 | 0.065 | 0.297 | 0.244 |
| | max likely exit | No | 0.782 | 0.074 | 0.480 | 0.183 | 0.877 | 0.062 | 0.357 | 0.210 |
| | surgical plan | Yes | 0.852 | 0.054 | 0.552 | 0.227 | 0.936 | 0.070 | 0.487 | 0.216 |

STN volume given by the respective surface mesh. The results show that MER-fitting effectively improves the localisation of the electrodes within the STN boundaries.

We used two different techniques to initialise the mutual position of the microelectrode trajectories and the STN surface before MER fitting: i) one technique uses the surgically planned target location, co-registered to the patient native space, and ii) the second method then shifts the depth 0 at the central electrode to the maximum likely entry point, according to the surface distribution described above (i.e. the max likely exit method). The initial position and the final position of the electrodes with respect to the STN containment are evaluated in Table 1 –the initial positions being identified as without the MER fit. Overall, the two initialisation techniques showed similar performance, while the maximum likely entry initialisation led to a marginally better result after MER fitting.

Another view can be provided by analysing the shifts introduced by the MER model with respect to the initial position, shown in Fig 9: the maximum likely exit initialisation (a) is compared to the aggregated shift resulting from the MER fitting (b), and the MER-fit, initialised using the surgical plan (c). All the methods result in similar transformation parameters with positive median shift in Ty and negative median shift in Tz and Tx being centered around zero due to the x-axis origin being aligned with the AC-PC line.

Overall, MER-based fitting showed its ability to markedly improve the localisation of the MER electrodes (Fig 8) with respect to the automatically segmented STN surface. Therefore, it may provide a viable method for intra-operative visualisation of the MER trajectories. This approach, however, has yet to be tested, and more detailed validation of the anatomical relevance of the final segmentation is yet to be carried out.

## IV. Discussion

We presented a novel model, combining image-based segmentation of subcortical structure with electrophysiology-based brain-shift correction, intended for intra-operative use. The following sections discuss results of the individual processing and evaluation steps.

### A. Manual labelling and image quality

The low score of manual relabelling can be explained by the low signal-to-noise ratio on the clinical images that were used. The overlap was lower in the case of the STN, compared to red nucleus, as it is relatively small in size and its boundaries are smooth and have relatively low contrast in the T2 images. The distribution of the overlap scores in manual images can be used to estimate the outlier fraction of the data for robust statistics, as we can expect that the error of our labels is no less than 1-$u$, where $u$ is the overlap coefficient of our relabelling procedure.

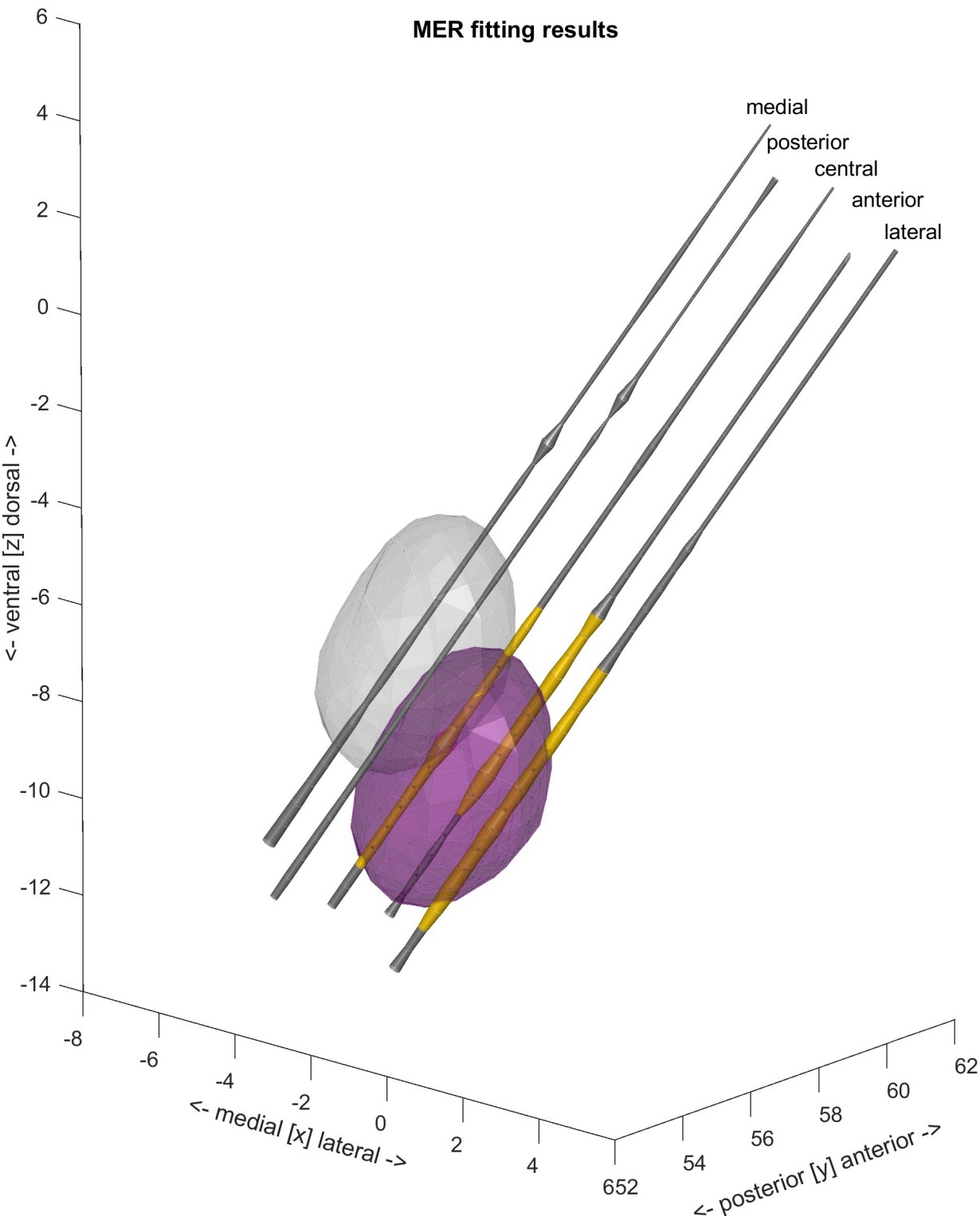

**Fig 9. Electrode positioning within STN initialised (grey) and fitted position (purple).** Individual 5 MER electrodes are represented by the cyllinders. Width of the cyllinders represents normalised MER signal energy (NRMS) at individual recording sites. MER sites, annotated as within STN are marked yellow, outside STN in grey. Units in milimeters in individual patient's native space.

For the purposes of estimating the distributions when estimating statistical shape model parameters in the learning stage, we set the number of outliers to 0.3 of all labels and considered them distributed as multivariate Gaussian.

## B. Image-based segmentation

The dice scores for segmentation of STN using the presented model on unseen data is comparable with the previously reported methods [7, 10]–see Table 2. The performance is similar to that in Visser's paper, although Visser's work used high-contrast 7T data of two modalities (T2 and quantitative susceptibility mapping), as opposed to the 1.5T clinical T2-weighted scans in work presented here. Also, the previous work by Visser compares the voxel-based overlap, whereas we calculate the overlap based on intersecting mesh volumes, meaning that a boundary correction method cannot be used. The segmentation results for other anatomical structures are also comparable. A closely related method by Patenaude [12] did not report results for the basal ganglia structures of interest.

A recent paper by Manjon [8] presented a patch-based technique called pBrains, which has very high accuracy segmentation–and the authors made their method available. We applied pBrains to our clinical dataset, and we obtained accuracy comparable with the accuracy of our method–which was very much lower than the accuracy reported in the paper (see Table 2). The inferior result on our dataset when our dataset was used can be explained by the image quality and by the fact that the initial model training had been on a different dataset. In addition, the pBrains model does not have shape priors for the STN shapes; it can therefore produce empty STN results in extreme cases.

## C. Post-surgery statistical analysis

The results of typical intensity profiles confirm the expectation that the intensities in the dorso-medial area grow most sharply–this is the area of the STN with a bright region close to the ventricle. The anterior area of the STN is then divided into two parts, represented by red and blue clusters; the red cluster is bounded by a high iron concentration on the SN, whereas the blue cluster covers the lateral part and partially includes the posterior STN area. Similarly, the posterior STN cluster is present in the same area has the same intensity representation as in the red and blue clusters. The sharp surge of intensity in the medial boundary of the STN could be explained by a dramatic rise in iron content in this region [36]. Most subjects have electrodes implanted across the posterior mid-section of the STN, which corresponded with clinically proven targets [11, 37].

## D. Full model with MER fitting

As can be seen from the results, the MER-based fitting improved the accuracy of correct containment of the MER recording sites by about 8%, compared to the initial conditions. This was

**Table 2. Overview of subcortical segmentation results (dice scores).**

| Method | Data | RN left | RN right | STN left | STN right | SN right | SN left |
|---|---|---|---|---|---|---|---|
| SSM–our method–mean (sd) | 1.5T (T2) | 0.749 (0.107) | 0.761 (0.104) | 0.641(0.134) | 0.622(0.115) | 0.643 (0.138) | 0.698 (0.147) |
| pBrains [8] | 3T (T2) | 0.930 | 0.908 | 0.872 | 0.878 | 0.925 | 0.908 |
| pBrains our data | 1.5T (T2) | 0.738 | 0.714 | 0.520 | 0.493 | 0.537 | 0.539 |
| CNN [42] | 3T (T1, T2) | - | - | 0.581 | 0.581 | - | - |
| NN [7] | 3T (T1, T2) | - | - | 0.671 | 0.671 | - | - |
| MIST [10] | 7T (T2,QSM) | ~0.852 | ~0.852 | ~0.62 | ~0.62 | ~0.752 | ~0.752 |

1 Derived from plots

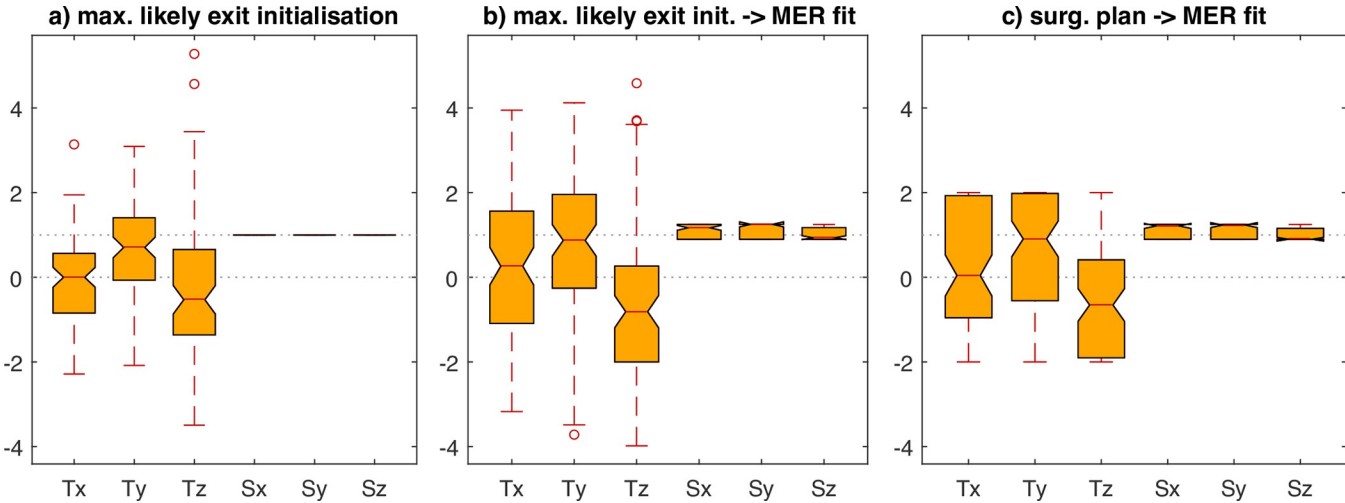

**Fig 10. Overview of transformation parameters, applied to the MRI-based automatic STN segmentation in the MER-fitting stage: Translations (Tx, Ty, Tz) and scalings (Sx, Sy, Sz).** The panel a) shows initial translation according to the maximum-likely entry and exit projected to patient native space, which is further refined by MER-fit in panel b) Panel c) shows MER-fit results initialized using surgical plan. Apparently, the overall character is retained, while eliminating outliers.

driven mostly by gains in specificity (correctly excluding the non-STN recording sites). The relatively low sensitivity (correctly including the STN recordings) in all cases may be caused by the electrical field of the STN being dislocated, compared to the T2 MRI artefact, as previously reported by Bus [38] and more recently by Oxenford et al. [39]. While the existing literature suggest a displacement of the MER-based and MRI-based STN, our results also suggest the electrophysiology-based STN to be larger. This is indicated by the scaling factors resulting from the MER-based fitting, which are often close to the pre-set constraint of 120% (see Fig 10), as well as by the low sensitivity of the method. While our model works with smooth representation of the STN boundaries both in the MRI image, as well as in the MER, the determination of the hard boundary of the STN is a design decision, bound directly to the size of the resulting STN. For this initial feasibility evaluation of the proposed method, we decided to keep the resulting STN size close to that of the size of the T2 MRI hypointensity based manual labels to limit the parametric space, at the sake of lower sensitivity. While future thorough evaluation of the overlap between T2 hypointensity and electrophysiological STN would be of high interest, higher quality MRI data would be needed to provide accurate estimates.

Overall, the accuracy of STN segmentation was markedly improved by the use of electrophysiology-based fitting. However, these accuracy results are inferior to several of the previously presented classification results based on individual recording sites or single trajectories, such as Moran [19]. This is likely due to the number of topographical constraints employed in our approach, which should provide a more anatomically relevant result at the expense of lower numeric accuracy. Using data from the same centre, a recent study by Khosravi [18] showed STN classification accuracy of about 92%. We assume that this is close to the accuracy of manual labelling of the MER data, and it is consistent with our own studies using data from other centres [40].

A similar task of combining microelectede recording with imaging data was approached in a recent study LEAD-OR [39], where the authors implement clustering of NRMS features and estimation of a brain shift with coregistration based segmentation approach with manual refinement. The LEAD-OR system focuses directly on intra-operative visualisation and assessment and integrates with a clinical microrecording system to visualise the NRMS of the

microelectrode recordings. While the authors also provide a coregistration-based segmentation of the subcortical structures, the authors did not provide segmentation based metrics as their pipeline included a manual refinementstep and therefore considered as a ground truth. The paper itself also does not mention the scanning properties and quality of imaging data which is necessary to assess accuracy of the method. Similarly to our work, the authors also utilise clustering approach with NRMS values which is then merged with segmented boundaries. Compared to our approach, the LEAD-OR authors provided a simpler model, relying more on human input and corrections. On the contrary, we propose a fully automated method for MRI-based segmentation and fusion with MER–which may be a valuable future component of a system similar to the LEAD-OR.

Regarding the impact of outliers, the max likely exit initialisation introduces large >3mm shifts on several occasions, which occurs due to a combination of nuisance factors, affecting the mapping of the max likely exit vertex to the individual segmented STN, as well as small co-registration inaccuracies. Due to this fact and as the max likely exit initialisation provides small benefit over the surgical plan, we consider the surgical-plan-based initialisation more anatomically accurate.

It should be noted that both initialisation techniques use only data available at the time of surgery and, therefore, can be used for intra-operative decision-making. The method may provide the surgical team with a 3D estimate of the localisation of the electrodes in the vicinity of an individually segmented STN, which provides more detail than the atlas-based fitting or simple sketching that are often used to depict the mutual position of the electrodes and the STN. This is especially true if the high anatomical variability of the STN is taken into consideration [41].

### E. Limitations

Our results yield several limitations, induced mostly by the dataset that is used. First, clinical 1.5 T MRI data provide relatively poor contrast among the structures of interest, and this is especially true for STN. This dramatically impacts the segmentation results, as is also documented by the relatively low overlap of the two series of manual labels in the relabelling procedure.

Second, the resulting localisation of the electrodes with respect to the STN could not be validated adequately because, in our experiments, the co-registration of the postoperative CT images to the patient native space was insufficiently accurate. We therefore were not able to compute validations accurately.

Third, all processing was done offline, long after the surgeries had been completed, and when complete, microelectrode exploration along all trajectories was available. In order for the solution to be usable intra-operatively, additional experiments with partial trajectory data need to be performed to simulate the advancement of the electrodes, and it is necessary to test the stability of the solution as more data are added. Concurrent evaluation of the MER data is typically also used to stop the recording when the ventral border of the STN or the SN is detected. This capability is yet to be added to the model.

### V. Conclusion

In this paper, we have proposed a combined segmentation approach to identify the STN, which utilises information from both MRI and microelectrode recordings, trained on manually labelled data. The model presents a novel combination of automatic fusion of image-based and electrophysiology-based segmentation: these two modalities have been treated separately to date, but they provide novel possibilities with great potential when they are combined.

When evaluated using an extensive set of clinical MRI and MER data, our method has proved that it has the ability to segment the basal ganglia structures and to improve the localisation of the electrodes within STN with an overall accuracy of 85%. When appropriately deployed, the model may provide an additional means for intra-operative decision making and also a tool for computing the surface distribution of image-based or electrophysiology-based statistics, thanks to the point correspondence. Our method also has the ability to perform segmentation and coregistration automatically on new subjects. In order to prove its clinical utility, however, it is necessary to carry out additional experiments with partial MER data and also to perform an anatomical validation of the resulting segmentation using higher-field-density MRI.

## Supporting information

**S1 File.**
(ZIP)

**S2 File.**
(ZIP)

## Acknowledgments

The authors are grateful to Dr. Mandar Jog from the Lawson Health Research Institute for providing the study data.

## Author Contributions

**Conceptualization:** Igor Varga, Eduard Bakstein.

**Data curation:** Greydon Gilmore, Jaromir May.

**Formal analysis:** Igor Varga.

**Investigation:** Igor Varga.

**Methodology:** Igor Varga, Eduard Bakstein.

**Supervision:** Eduard Bakstein, Daniel Novak.

**Validation:** Jaromir May.

**Visualization:** Igor Varga, Eduard Bakstein.

**Writing – original draft:** Igor Varga, Eduard Bakstein, Daniel Novak.

**Writing – review & editing:** Igor Varga, Eduard Bakstein, Daniel Novak.

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
