## [Decision Letter · Decision Letter 0]

16 Oct 2023

PONE-D-23-24115Statistical Segmentation Model for Accurate Electrode Positioning in Parkinson's Deep Brain Stimulation Based on Clinical Low-Resolution Image Data and ElectrophysiologyPLOS ONE

Dear Dr. Varga,

Thank you for submitting your manuscript to PLOS ONE. After careful consideration, we feel that it has merit but does not fully meet PLOS ONE’s publication criteria as it currently stands. Therefore, we invite you to submit a revised version of the manuscript that addresses the points raised during the review process.

Dear Dr. Varga et al.,

We appreciate the time and effort invested in contributing your research to PLOS ONE. After a thorough evaluation, two expert reviewers found several points of merit, notably that your model is predicated on 1.5 T imaging and incorporates microelectrode information and would provide useful information and insight to the field.

However, the reviewers have also provided important feedback that suggests several areas requiring significant revisions. These points are critical to enhance the quality and impact of your work. 

Please see below for reviewer feedback.

Thank you,

John A. Thompson

We look forward to receiving your revised manuscript.

Kind regards,

John A. Thompson

Academic Editor

PLOS ONE

“The study was supported by the grant of the Czech Ministry of Health, grant No. NV19-04-00233 (project CLIMABI), by the Research Centre for Informatics, grant No. CZ.02.1.01/0.0/16_019/0000765 and by the students' grant of the Czech Technical University: Biomedical data acquisition, processing and visualisation, number 22∕165∕3∕3∕13.”

Reviewers' comments:

Reviewer's Responses to Questions

**Comments to the Author**

1. Is the manuscript technically sound, and do the data support the conclusions?

Reviewer #1: Yes

Reviewer #2: Yes

2. Has the statistical analysis been performed appropriately and rigorously? 

Reviewer #1: Yes

Reviewer #2: Yes

3. Have the authors made all data underlying the findings in their manuscript fully available?

Reviewer #1: Yes

Reviewer #2: Yes

4. Is the manuscript presented in an intelligible fashion and written in standard English?

Reviewer #1: Yes

Reviewer #2: Yes

5. Review Comments to the Author

Reviewer #1: In this study, the authors developed and tested software to automatically map the borders of deep brain nuclei including STN, automatically classify MER recordings as inside or outside STN, and correct for brain shift that results in mismatch between imaging and MER. They found 60% overlap between manual and automatic MRI segmentations and 85% accuracy between MER labeling and final STN modeling.

Main comments:

- The authors have taken on a large task by attempting to develop both software for anatomic segmentation and MER fitting. A majority of the manuscript is dedicated to automatic STN segmentation which has been explored at length including using other methods which the authors also mention. The more interesting aspect is the classification and fitting of MER data to imaging which receives less attention in the manuscript.

- The low sensitivity of the method is concerning. A relatively high specificity is to be expected given that most recording sites are not in STN.

- As the authors acknowledge, the data set is limited by the low field strength (1.5 T) of the MR images and the inability to coregister postoperative CT imaging with preoperative MRI. This to some degree reflects the time at which the data were collected (2008-09). This does limit the interpretation of the results given that the data are of lower quality than current clinical data (generally using 3T MRI).

- The authors do not discuss other authors that have attempted imaging-MER coregistration, particularly the creators of the LEAD-OR system (PMID: 35594135).

Minor comments:

- The age range is listed as 38-75 years, and later the inclusion criteria are listed as 45-85 years; these are not compatible

Typographical comments as requested by the journal:

- abstract sentence 1 - "neuropsychiatric disorders" should read "neurologic disorders"

- abstract sentence 2 - "suppress brain shift" should read "compensate for brain shift"

- introduction, paragraph 7 - "injecting multiple microelectrodes" should read "inserting multiple microelectrodes"

- methods, data subsection, paragraph 1 - please use past tense rather than a mixture of past and present tense

Reviewer #2: In this manuscript, Varga and colleagues report a novel segmentation model used to automatically define basal ganglia structures relevant to deep brain stimulation surgery. Using a leave-one-out validation, they demonstrate that the overlap between the automatically segmented nuclei and manually segmented nuclei is approximately 0.62 – 0.64 for the STN. Helpfully, this is compared to other reported models. Although other models have reported higher overlap (Dice) scores, there is certainly an element of clinical applicability here, as the model uses only 1.5T MRI images, whereas others have used higher intensity MRI (3T or 7T). Next, electrophysiology is incorporated into the model in the form of microelectrode recording data. The authors demonstrate that this strategy yields an overall accuracy of 85% for identifying the subthalamic nucleus, compared to the gold standard of a clinician-determined identification of location. The methods used to generate and test the model are rigorous and adequately explained. As such, this manuscript would likely make a novel contribution to the literature. I would suggest the following revisions:

Manuscript

-Introduction: “Manual identification of STN and SN location from MRI images is the most common way to identify the most appropriate DBS target coordinates”. Either a citation should be provided for this statement, or the statement should be altered to include the use of indirect targeting, which is commonly employed for stereotactic surgery for PD.

-Methods: Please include state the time between surgical implantation and the postoperative CT scan, as variation in this could lead to variability in the presence of intracranial air or other factors which could affect brain shift.

-Results and Discussion, Section C. Several methodological steps not discussed previously are presented in this section. For example, the cluster analysis used to generate Figure 7 and the evaluation of electrode entry and exit points with respect to the STN are discussed here (and also in the figure legend for figure 8). These should be presented in the Methods section.

-In general, I would suggest separating the “Results and Discussion” section into separate “Results” and “Discussion” sections, in order to better clarify what is a result of the present analysis, and what is the authors’ interpretation of these results. Discussion of and comparison to other papers in this field should also be moved to the Discussion section.

Figures and Tables

-Figure 1 is somewhat confusing in its layout. It purports to show the evaluation stage for the trained model, but it is not clear to me that this is showing an evaluation method for the model. Instead, it seems to also show a schematic of the model’s inputs and outputs. I would suggest re-formatting this image to either more clearly show some sort of evaluation method or convert it fully to a schematic depicting how the model operates.

-Figure 5: Abbreviations used on the x-axis should be defined in the figure legend. The y-axis needs a label and definition of the units. The figure legend states that the STN was joined with SN for the segmentation, but the figure shows separate segmentation results for the STN and SN; this needs to be clarified.

-Table 1: Recommend adding interquartile ranges to the Dice scores reported with the SSM, to correlate with Figure 5. Abbreviations used in the table should be defined in the table legend. Also, the numbers 1 and 2 are used as annotations for footnotes, but do not correspond to any data marked “1” or “2” in the table.

-There are two tables labeled as “Table 1”. The second one (i.e., Table 2) is not formatted well and overlaps with the body of the text.

-Figures containing 3 dimensional representations of the STN (Figures 7, 8, 9) should have consistent axis labels and some description of the units. For example, the y-axis of Figure 8 ranges from 17-26, and the y-axis of Figure 9 ranges from 52-62. What are the units?

References

-Some references appear to be incomplete (e.g., missing journal information). These include references #24, 30. Reference 26 provides an archive website, but the original journal of publication is not cited.

Minor issues

-Introduction: The reference for Zwirner et al., 2017 is provided in a different style from other references (i.e., all other references are numbered in the text).

-Discussion: avoid using abbreviation “w.r.t” without first defining it.

-Discussion, Section D, fourth paragraph: “ectrodes” should be “electrodes”

-Figure 10: The figure caption appears to be cut off.

-Limitations: There is a sentence which is cut off or incomplete (“We therefore stick to the”)

6. PLOS authors have the option to publish the peer review history of their article (what does this mean?). If published, this will include your full peer review and any attached files.

Reviewer #1: No

Reviewer #2: No

---

## [Author Response · Author response to Decision Letter 0]

5 Dec 2023

Dear Reviewers,

Thank you for your valuable input. We put a lot of effort to edit the manuscript according to your suggestions.

Please check responses to your comments in the responses_to_reviewers_and_editors.docx document.

Thank you,

Varga Igor

---

## [Decision Letter · Decision Letter 1]

23 Jan 2024

Statistical Segmentation Model for Accurate Electrode Positioning in Parkinson's Deep Brain Stimulation Based on Clinical Low-Resolution Image Data and Electrophysiology

PONE-D-23-24115R1

Dear Dr. Varga,

We’re pleased to inform you that your manuscript has been judged scientifically suitable for publication and will be formally accepted for publication once it meets all outstanding technical requirements.

Kind regards,

John A. Thompson

Academic Editor

PLOS ONE

Additional Editor Comments (optional):

I hope this email finds you well. It is with great pleasure that I inform you of the recent decision regarding your manuscript, titled "Statistical Segmentation Model for Accurate Electrode Positioning in Parkinson's Deep Brain Stimulation Based on Clinical Low-Resolution Image Data and Electrophysiology" submitted to PLOS ONE.

I am delighted to notify you that your manuscript has been thoroughly reviewed and has now been officially accepted for publication in our journal. The reviewers have provided positive feedback, and we believe your work will make a significant contribution to the field.

Best,

John

Reviewers' comments:

Reviewer's Responses to Questions

**Comments to the Author**

1. If the authors have adequately addressed your comments raised in a previous round of review and you feel that this manuscript is now acceptable for publication, you may indicate that here to bypass the “Comments to the Author” section, enter your conflict of interest statement in the “Confidential to Editor” section, and submit your "Accept" recommendation.

Reviewer #1: All comments have been addressed

Reviewer #2: All comments have been addressed

2. Is the manuscript technically sound, and do the data support the conclusions?

Reviewer #1: Yes

Reviewer #2: Yes

3. Has the statistical analysis been performed appropriately and rigorously? 

Reviewer #1: Yes

Reviewer #2: Yes

4. Have the authors made all data underlying the findings in their manuscript fully available?

Reviewer #1: Yes

Reviewer #2: Yes

5. Is the manuscript presented in an intelligible fashion and written in standard English?

Reviewer #1: Yes

Reviewer #2: Yes

6. Review Comments to the Author

Reviewer #1: (No Response)

Reviewer #2: (No Response)

7. PLOS authors have the option to publish the peer review history of their article (what does this mean?). If published, this will include your full peer review and any attached files.

Reviewer #1: No

Reviewer #2: No

---

## [Editor Report · Acceptance letter]

12 Feb 2024

PONE-D-23-24115R1 

PLOS ONE

Dear Dr. Varga, 

I'm pleased to inform you that your manuscript has been deemed suitable for publication in PLOS ONE. Congratulations! Your manuscript is now being handed over to our production team.

Kind regards, 

on behalf of

Dr. John A. Thompson 

Academic Editor

PLOS ONE